# Anti-Soiling Coatings for Enhancement of PV Panel Performance in Desert Environment: A Critical Review and Market Overview

**DOI:** 10.3390/ma15207139

**Published:** 2022-10-13

**Authors:** Mohammad Istiaque Hossain, Adnan Ali, Veronica Bermudez Benito, Benjamin Figgis, Brahim Aïssa

**Affiliations:** Qatar Environment and Energy Research Institute (QEERI), Hamad Bin Khalifa University (HBKU), Qatar Foundation, Doha P.O. Box 34110, Qatar

**Keywords:** soiling, anti-soiling coatings, mitigation, hydrophobic, hydrophilic, outdoor testing

## Abstract

Areas with abundant sunlight, such as the Middle East and North Africa (MENA), are optimal for photovoltaic (PV) power generation. However, the average power loss of photovoltaic modules caused by dust accumulation is extreme and may reach 1%/day, necessitating frequent cleaning which adds to the cost of operations and maintenance. One of the solutions to the problem of PV soiling is to develop anti-soil coatings, where hydrophilic or hydrophobic coatings with spectral characteristics suitable for PV applications are added to the outer layer of PV glass. However, the effectiveness of such coatings depends extensively on climatic conditions and geographical locations. Since coatings add to the cost of solar panels, it is imperative that they are first tested for suitability at the intended location and/or in similar weather conditions prior to their large-scale deployment. This critical review focuses on various anti-dust technologies employed to mitigate the PV soiling issue. The in-depth comparison of the various developed techniques and materials aims at providing a relevant input in adapting the right technology based on particles’ accumulation mechanism, weather conditions, and geographical location. Though the mechanical cleaning process is the most used solution to date, development of thin film anti-dust coating could be a better alternative—when it is relevant—due to its abrasion-free capability, large deployment, economic viability, and durability. This review aims at serving as a reference in this topic, thereby paving the way to adapting efficient anti-dust coatings, especially in the MENA region and/or desert environment at large, where it is the most relevant.

## 1. Introduction

Photovoltaic (PV) systems inevitably depend on the Sun’s irradiance as well as spectral content along with component performances. On the other hand, installation area and atmospheric conditions also play a vital role in the output power to alter system performance. One of the issues is dust accumulation on PV panels, which has been underestimated, but can lead to a deteriorating factor for PV plants through limiting output power [1]. In the Middle East and North Africa (MENA) region, dust accumulation combined with moisture is an intrinsic phenomenon, hence, it keeps the scientific as well as engineering windows open to tackle the issues [1,2]. Many scientific works have been carried out for the last 30 years to bring the PV system into a steady state with reasonable and reliable mechanical as well electrical components [1,2,3,4,5,6,7,8,9,10,11,12,13,14,15]. However, dust accumulation on the surface while interacting with light can easily disrupt the planned function by reducing the power output, which can lead to complete system performance termination.

In general, dust is a matter less than 500 nm in diameter and composed of organic minerals [1,16]. Atmospheric dust can be due to soil floatation by the wind, volcanic eruption, and environmental contamination. The dust particle composition is not same in different places and can vary drastically from place to place [1]. Moreover, the accumulation character and rates also differ significantly in different locations. To date, the most established way of cleaning PV modules is mechanically using water or cleaning substances. However, such cleaning requires an intensive supply of water along with workforce, and finally, can lead to a very costly operational and maintenance system. In some previous works, misleading results of dust accumulation rate were reported without any consideration of dust density and particle characterization. As reported, a tilt angle of 30° may result in only 1% solar spectrum whereas 5% loss has been reported for a tilt of 0° to 50° [1].

Dust can be a major concern with intermittent natural rainfall even with a lower soiling rate on the panel surface. The degradation rate from soiling can be up to 14% due to unstable dusty weather conditions with a significant loss of transmittance of sunlight irradiance of 12%. However, this is not the case in the MENA region and natural rainfall is trivial [1,17]. There are four identified soiling mechanisms, which are (i) water-soluble salt cementation, (ii) organic material deposition, (iii) tension between surface and dust particles, and (iv) particles’ energy. The cementation mechanism is one of the main deteriorating factors as a large amount of dust and moisture levels can result in gluing of dust on PV panels [17]. This mechanism results in accumulation of cemented particles once dried (Figure 1). Organic layer deposition on the surface results in cleaning of the deposited salt being more difficult with a complex cleaning procedure.

Among the mitigation techniques, restoration has been utilized to bring back the original surface area by removing dust. Other techniques involve a repulsion mechanism of dust on surface, cleaning with water or with stronger chemicals and mechanical cleaning agents, and by tuning surface properties [18,19]. In addition, mechanical removal of soiling is considered as a logical strategy which involves the wiping of the surface with a cloth [1]. However, this technique requires a huge workforce along with process repetition. Automation using robotics can be used as well with chemical agents based on a mild detergent solution. Still, such techniques leave out the scope of surface degradation due to mechanical cleaning procedures. A balanced cost-effective process of surface cleaning requires less labor plus cleaning solvents and sufficient cleaning to reduce any power loss. Though water has been considered as one of the most effective washing agents, water scarcity is an issue in many regions of the world, including MENA. As studied previously, water as well as other chemical solutions are effective to remove dust particles [20,21]. Though the dust condensation is relatively higher in the MENA region, moisture effects take place as studied previously [13,14,15].

It becomes significantly important to develop a coating with desirable properties to unsettle dust from the surface with durability as well as reliability. An anti-dust surface coating has to be highly economic for large-scale application with the maximum durability under UV and infrared solar spectra [22,23,24]. Developing an anti-dust coating to repel the dust particles from the glass surface without disturbing any optical properties is not new. In the case of photovoltaic applications, various chemical agents such as polyvinyl chloride (PVC), polyethylene, and acrylics have been used to grow films on glass substrates. However, such coatings have been found to become unstable with age due to the organic molecules. A hydrophobic surface and lower surface tension have been identified as other important parameters to reduce dust accumulation on PV panels (Figure 2) [23,24]. This mechanism replaces sodium and potassium ions with aluminum ions. This molecular engineering changes surface properties from hydrophilic to hydrophobic. Additionally, aluminum improves water resistivity and surface durability. Ionic cross-linkers such as acrylics can replace aluminum ions due to better surface durability and hydrophobic behavior [25,26]. Generally, hydrophilicity can be formulated using high surface energy materials to resist dust accumulation and, due to soluble properties, such layers become dominant environmentally friendly materials. On the other hand, hydrophobicity happens with a high angle between water droplets and the surface, where gravity helps them to slide down the accumulated dust on the surface. Additionally, charge build-up from dust can be prevented using high surface conductivity materials which act as anti-static layers [25,26,27].

Water-repellent coatings have also been demonstrated as anti-dust and anti-moisture coatings [25,26,27,28,29,30]. In previous works, texturing the surface with different geometries has been developed to mitigate the soiling issue. Though super-hydrophobic materials have been tested as anti-dust coatings, they have been found to be unstable due to the nature of organic properties under the UV light spectrum.

PV performance drops drastically due to dust and a reduction in the solar spectrum has been measured from 20% to 50%, which eventually disturbs the overall performance of a PV system by a reduction up to 30% even with considerable dust accumulation. Soiling on the surface also increases reflection of concentrating solar power (CSP) systems, leading to a drastic drop in output power. It is one of the most degrading factors for reflective surfaces [31,32]. Generally, CSP plants are very sensitive to dust (much more than PCV plants) and require a high level of maintenance to mitigate the soiling issue. Both passive and active approaches have been adapted in order to prevent as well as repel the dust particles. Such technologies have been defined as state-of-the-art with an active engagement of physicists, chemists, and material scientists [1]. Super-hydrophilic or super-hydrophobic surfaces have been launched effectively for both wet and dry surfaces though further investigations are required to validate the lifetime of such films. The space industry has substantially adapted the repelling technology and major approaches have been evaluated to test the effectiveness with techno-economical studies. The best anti-dust coating must fulfill few criteria before a full launch such as being highly transparent in the visible range, having a longer lifespan, lower costs for mass production, and being non-toxic and feasible for large-scale fabrication.

## 2. Anti-Dust Coating

The dust accumulation on PV glass modules significantly reduces modules’ output power (Figure 3). Hence, the PV industry has been active in developing anti-soiling and anti-reflection coatings [33,34,35,36,37,38,39,40,41,42]. However, very few experimental works have reported such layers for a longer period of testing time considering natural sandstorms along with other cleaning procedures [43].

The loss in performance from PV soiling (i.e., dust accumulation) or ambient particulate matter (PM) effects depends strongly on the location with both depending on various parameters such as wind speed, density of dust particles, composition, etc. [45,46]. Soiling is a result of dust particles and the composition of such particles varies significantly around the world (Figure 4). The main component of the chemical composition strongly depends on localized and unstable sources [43].

In general, the soiling chemistry involves the composition of silica and metal oxide. In addition, other defective sources include salt, sulfuric acid molecules, and soot where sulfuric acid molecules originate by transformation from gaseous phase to solid phase [47,48]. Additionally, biofilms can grow on solar glasses [49,50]. Data analysis can be used as a smart measurement technique to quantify dust particles as it is important to conduct a survey on different areas before launching a large-scale PV plant. Overall PV performance can drop by 70% based on location due to the soiling effect [51,52]. Hence, it becomes obvious to make PV module cleaning operational through different techniques such as manual, mechanical, and robotic cleaning techniques. Robotic cleaning can be defined as an automatic or semi-automatic technique. Though it is quite difficult to eliminate the effects of dust fully, repelling dust can be carried out effectively using such techniques.

To date, no mitigation approach has been initiated in a systematic way to tackle soiling problems considering water consumption and labor force requirements [48,50,53,54]. Generally, a cleaning procedure only takes place when it becomes economically viable through the calculation of power generation vs. soiling effect (Figure 5 and Figure 6). Some places do not require advance cleaning techniques due to the natural rainfall which washes away any accumulated dust [55]. During dry periods in the southwest United States, cleaning takes place 1–2 times annually. On the other hand, in the MENA region, water scarcity is a dominant factor, hence, dry cleaning with a soft brush is commonly used. Besides this, robotic techniques are also being used to clean solar modules. Front glass standardization has been utilized for quite a long time for PV panels, which can be adapted accordingly to choose the correct cleaning procedure. Though surface coatings have been developed as anti-dust coatings, standard aging tests have not been performed yet. Many companies have developed a solution to include both anti-dust and anti-reflection coatings, which eventually improves the performance by 3% through better light management [56,57]. There has been a significant amount of effort to develop a functionalized surface layer as an anti-dust coating. Further investigations are in progress to understand the durability of such layers from the economical perspective under various conditions [56,57].

Anti-dust coatings that are highly transparent, anti-reflective, durable, and non-toxic with self-cleaning character are defined as the “Holy Grail” by the research community [44].

## 3. Mechanisms for Preventing Soiling by Hydrophobic and Hydrophilic Coating

A hydrophobic surface allows water to build up on the module glass surface and to utilize the kinetic energy to clean up the accumulated dust particles, whereas a hydrophilic surface allows spreading of the water molecules to wash away the dust particles. Choosing the right surface to prevent soiling depends entirely on environmental factors. Hydrophobic surfaces have gained enormous attention due to the capability of repelling dust particles along with organic molecules in drier regions, whereas hydrophilic surfaces have been used in areas of higher rainfall. In both cases, such surfaces need to satisfy the criteria of stability under UV light, greater durability in harsh environments, and thin films (<500 nm) to be used as anti-reflection layers. Generally, anti-reflection coating technology uses silica, silica nanomaterials, metal oxide, and metal fluoride. Other organic materials consist of PET, PMMA, and PDMS [59]. There are many well-established technological developments to fabricate such layers on substrate, which consist of stacking mixed refractive index materials, gradient refractive index films, surface texturing, and introducing porosity [59,60].

The industrial deployment of anti-soiling coatings has not reached maximum capacity as the cleaning period is trivial (Figure 7). In this case, passive cleaning has offered the highest level of promise [61]. A reduction in soiling up to 80% has been reported for a shorter period, whereas, for a longer period, the rate is between 20% and 50%. This depends strongly on environmental conditions as well as the type of coating [53,54,55,62].

A detailed reliability test of silica coating as anti-dust coating was reported by Kensuke [59] for PV modules. As reported, a loss in solar spectrum happens mainly due to the lack of anti-reflection and anti-soiling coating. This suggests an obvious development of thin film coatings with dual functionalities such as better light management within PV modules by reducing reflection and mitigating soiling issues. In this regard, CIGS-based PV panels were developed with a front transparent layer with silica coatings with an optimized thickness and refractive index. Such PV devices were exposed to an outdoor environment and reliability was tested. As a measure, output performance improved by 3.9%. The aging test was also performed for 3.5 years without any solid degradation and later the same optimized layer was used for Si PV modules with the same performance enhancement.

## 4. Silica Coating on Module by Dipping/Immersing

In this technique, a porous silica layer forms on the glass panel by evaporation of the solvent. This silica layer consists of many hydroxyl groups, which helps in adsorbing water molecules on the surface [39]. Figure 8 shows a simple coating process based on sponge phase resin and the surface of the PV panels after coating [57]. Figure 9 shows the surface of different PV modules (based on Si Technology) where two modules were coated and the other left uncoated to act as references. In Figure 10, we can see the normalized daily performance ratio (PR) of arrays B (without coating, called here NPRB) and C (with coating, called here NPRC), twice normalized daily PR of arrays B and C (NPRB and NPRC were divided by the normalized daily PR of reference array, and the ratio of NPRC/NPRB. One may note that the ratio of the NPR of the coated on the uncoated module is above 1, which is a clear indication of the added value of the coating on the module performance [57].

In a study by Wang et al. [63], controlling oxygen flowrate during the growth of silica films could result in altering the microstructure using ion beam sputtering. The films were studied for optical parameters and morphology. As characterized, refractive index values change with the oxygen flowrate, where they increase for the highest oxygen flowrate. In another study [64], super-hydrophobicity and super-hydrophilicity were studied for nanostructured coatings. The super-hydrophilic surface was highly reflective with a contact angle of 0° and for the super-hydrophobic surface, it was 165°. Growth of nanostructured silica nanoparticles on a mirror demonstrated super-hydrophilic behavior due to the rough surface and another sample with lower surface energy ligand molecules confirmed super-hydrophobic surface properties. Both layers showed a minimal reflection loss with excellent dust repelling properties due to the engineered roughness that reduced the adhesion forces of dust particles. Outdoor testing showed that the coating of such layers outperforms the uncoated glass layers due to the ability of self-cleaning. However, the tests conducted for the longer period of time confirm that the super-hydrophilic surface was less affected by the cementation process of organic hard water molecules, offering the highest level of advantage. Thus, super-hydrophilic layers show potential benefit when used for solar energy harvesting applications. Figure 11 shows highly reflective, anti-soiling nanoparticle texture solar mirrors with no coating, with super-hydrophilic (SPH) coating, and with super-hydrophobic (SP) coating along with the water contact angle measurements on the coated mirrors and SEM micrographs [62]. Table 1 summarizes the chemical composition of SP and SPH coating determined by XPS analysis [62].

## 5. Adhesion Force Reduction in the SHP Coating

The van der Waals force between the dust particle and surface decreases for higher surface roughness for both super-hydrophilic and super-hydrophobic surfaces. The theoretical calculation shows that SPH surface adhesion is higher than that of SP surfaces by 30% whereas experimental measurement confirms that the force is slightly lower than that of SP surfaces. As confirmed by both techniques, surface roughness plays a vital role in determining the relation between adhesion force and the coated substrate. Figure 12 and Figure 13 show the adhesive force of spherical dust particles deposited on SPH-coated, SP-coated, and uncoated mirrors as a function of relative humidity [62], and the measured vs. calculated adhesion forces between a silica sphere and SP and SPH surfaces [62], respectively, along with AFM surface morphology analysis of the uncoated surface, the SPH, and SP coated surfaces [62].

## 6. Anti-Soiling and Self-Cleaning Performance of SPH Coated Mirror

The super-hydrophilic layers on mirrors show an excellent self-cleaning property. As shown in Figure 14, on a mirror with a 45° tilting angle with half coated and the other half uncoated, dust particles rarely scattered on the coated surface whereas large agglomerations with densely packed dust particles were found on the uncoated surface. An artificial simulation was carried out to replicate the original air and it was found that the surface was cleaned effectively by removing the loosely bonded dust particles. The development of non-porous-textured silica oxide layers showed excellent self-cleaning properties due to super-hydrophilicity. The wetting behavior of such layers resulted in reducing surface adhesion energy. Compared to the super-hydrophobic surface, the performance of dust reduction was 2.5 times higher and was found to be due to reducing organic cementation. In general, a super-hydrophilic surface can be proved to be hydrophilic through texturing of the surface and such morphology strongly depends on background gas pressure [39].

For uniform surface,
Cos θ = (𝜇_1_ − 𝜇_2_)/𝜇_3,_(1)
𝜇_3_ × Cos θ = (𝜇_1_ − 𝜇_2_).(2)

With texturing,
𝜇_3_ × Cos θ = φ (𝜇_1_ − 𝜇_2_) − (1 − φ) 𝜇_3_,(3)
where φ stands for the insignificant area covered by the texturing on the surface, θ is the angle of the droplet, 𝜇_3_ is the tension between liquid and vapor, 𝜇_1_ is the tension between surface and vapor, 𝜇_2_ is the tension between surface and liquid. This mechanism is well established and known as the theory of Wenzel and Cassie-Baxter [39]. For a super-hydrophobic surface, a water droplet rolls down the dust particles using gravitational force. Among all types of materials, SiO_2_ is the most dominant anti-soiling coating due to its empirical properties. Table 2 summarizes the main achievements in the development of TiO_2_-based anti-reflective coatings (ARCs) and figures of merit showing the resulting enhancement in solar cells’ performance.

To date, various approaches have been investigated to improve the durability of super-hydrophobic surfaces. Such approaches include the growth of SiO_2_ layers with nanoparticles, low surface energy substances, polymers, intertwined cellulose/SiO_2_ layers, and optimized micro/nanohierarchical structure [72,73,74].

In some parts of the world, snow can act as a degrading factor for output performance of photovoltaic panels. It is considered as a highly reflective material, which can reduce the incident solar radiation and may result in performance drops of the photovoltaic panels. Both super-hydrophobic and super-hydrophilic surfaces have been suggested as solutions to ice formation along with surface heating, however, to date no solid ice-phobic surfaces have been demonstrated.

## 7. Solar Panel Coatings Market Overview

Clean energy from PV modules has attracted the highest level of attention especially in areas with water scarcity. Even though various technologies are available in the current PV market, performance is significantly influenced by the reflection loss and soiling issues. Thus, various approaches have been established to develop thin films with various functionalities such as anti-reflection, anti-soiling, anti-fogging, etc. Figure 15 shows the global solar panel coating market [75]. The market of worldwide PV coating technology is estimated to reach around ~ USD 2318 million by 2026, which is higher than the market of ~ USD 1500 million in 2020. The main drive for such development is a joint venture of private and public companies to fulfill the need for sustainability.

The market research is driven by four important properties: anti-reflection, self-cleaning, anti-soiling, and anti-abrasion. Anti-reflection coating development is increasing rapidly to improve light management, thus improving overall device performance. Hydrophobic surface coating is projected to gain attention in areas at higher elevations since it does not allow water to remain on the surface. Figure 16 shows the solar panel coatings market segmentation analysis [76].

Currently, some manufacturers are active in developing large-scale production of thin film coatings, such as:− Arkema;− Fenzi SpA;− NanoTech Products;− Koninklijke DSM;− PPG Industries;− Unelko Corporation.

There is a huge demand for expansion of the market due to the initiatives and incentives provided by various governments. The market of PV panel coatings is almost proportional to the production of PV modules. The drive for renewable energy is significant due to the concern over environmental changes. Asia Pacific held the highest market share in the global market of 2017. The market is expected to increase exponentially in the coming years, followed by North America and Europe. The projected period of expansion in Asia Pacific is between 2017 and 2026, considering rooftop solar module installation in both residential and commercial spaces. Table A1 in the Appendix A summarizes the main ADC market-dominating commercial companies.

## 8. Conclusions

PV soiling can significantly reduce the energy yield of the modules, especially in desert climates due to high airborne dust concentration, frequent dust storms, and rare rain events. The current vast deployment of PVs in deserts requires adequate and economically feasible strategies for the mitigation of soiling. In this regard, anti-soiling coatings (ASCs) could be a promising solution as they could significantly reduce the accumulation of dust and resultant soiling losses. The investigated results indicate huge variations of anti-soiling functionality of the coatings depending on exposure period and location. Overall, both improved and decreased anti-soiling performances have been observed for coatings compared with an uncoated reference glass. An investigation on the correlation between the daily soiling loss and weather parameters indicates that besides precipitation and particulate concentration in the air, dew and its interaction with the coatings is also an important influencing parameter strongly affecting soiling rates. Looking at the two most promising coatings with very low adhesion properties, the overall performance benefit is higher for a hydrophilic surface in highly humid areas (coastal) and a hydrophobic surface in dry areas (central desert). We have also captured the available techniques in the market to mitigate the soiling issue in terms of growth process, cost analysis, large-scale production, and material selection. This comprehensive review shows that various techniques and materials are available to be adapted based on the specific needs. Currently, both active and passive cleaning techniques are suitable to be used for anti-soiling functions. However, the market niche is yet to develop suitable coatings, which will have multiple functions such as anti-reflection and anti-soiling.

## Figures and Tables

**Figure 1 materials-15-07139-f001:**
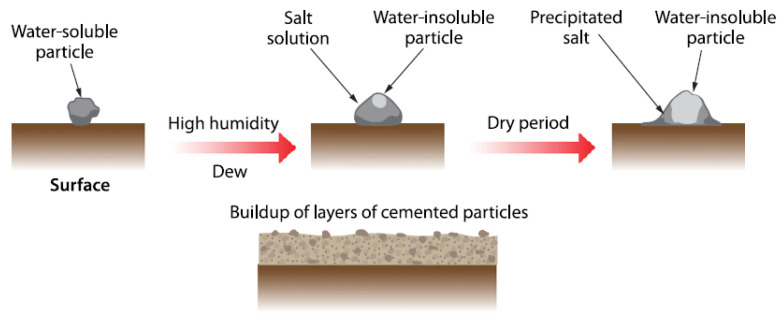
Dust–moisture (water) cementation process.

**Figure 2 materials-15-07139-f002:**
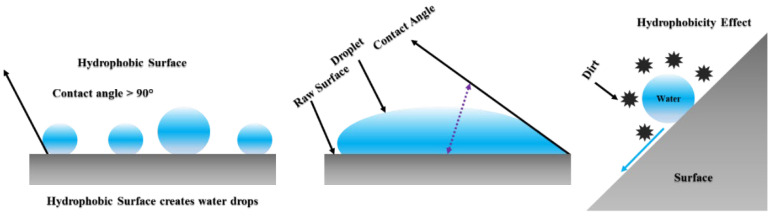
Hydrophobic surface as anti-dust coating.

**Figure 3 materials-15-07139-f003:**
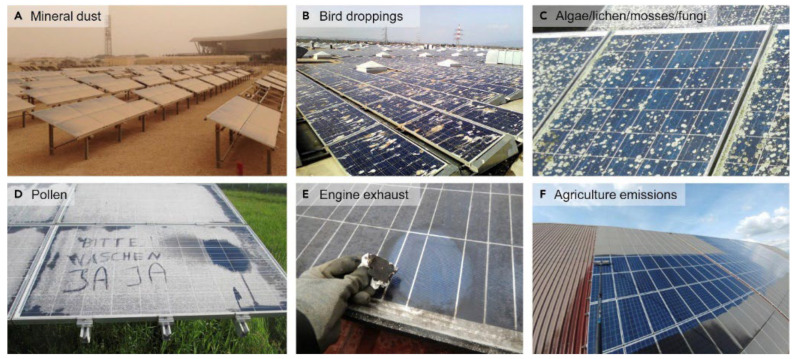
Examples of soiling: Overview of different soiling types with exemplary photographs of soiling by (**A**) mineral dust in a desert area, (**B**) bird droppings, (**C**) algae, lichen, mosses, or fungi and (**D**) pollen in wet and moderate climates, (**E**) engine exhaust from an industrial area, and (**F**) agricultural emissions. Reproduced with permission from Ref. [44].

**Figure 4 materials-15-07139-f004:**
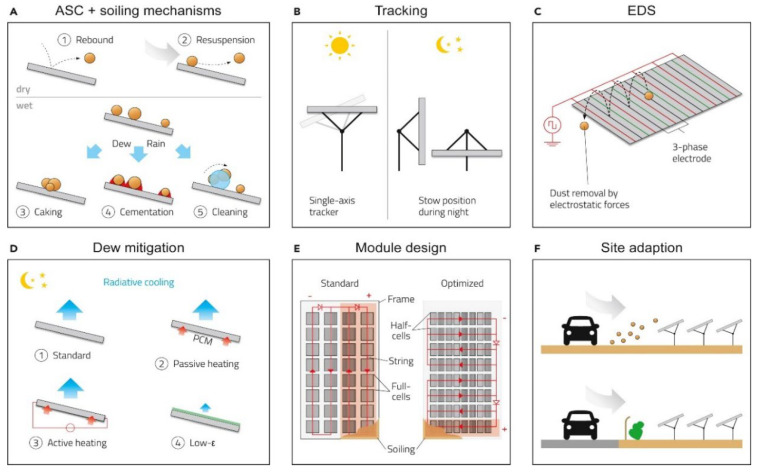
Schematic illustration of soiling mitigation technologies: (**A**) Important soiling mechanisms which could be addressed by anti-soiling coatings (ASCs). (**B**) Single-axis tracking and optimization of night stowing position. (**C**) Working principle of EDS (standing wave version). (**D**) Dew mitigation by low-ε coatings and active and passive heating. (**E**) PV module design approaches for soiling loss reduction: the red overlay indicates lost cell strings due to soiling. (**F**) Site adaption. Reproduced with permission from Ref. [44].

**Figure 5 materials-15-07139-f005:**
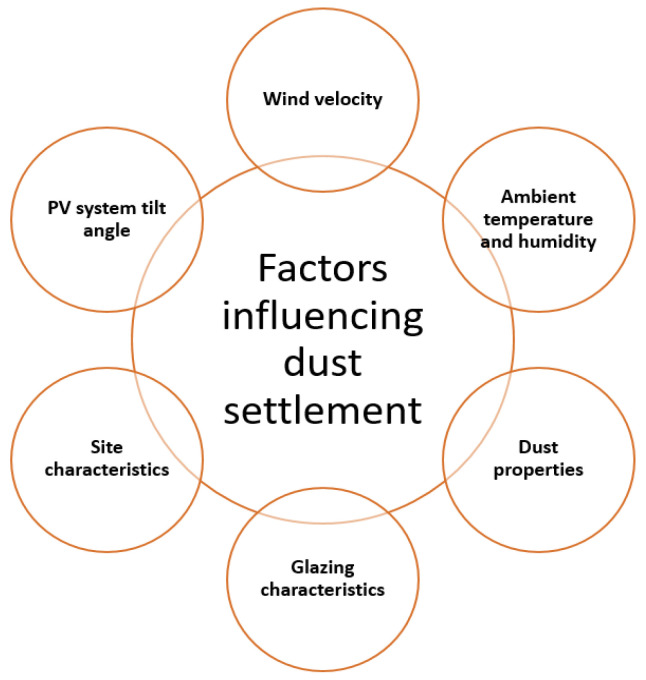
Factors influencing dust settlement.

**Figure 6 materials-15-07139-f006:**
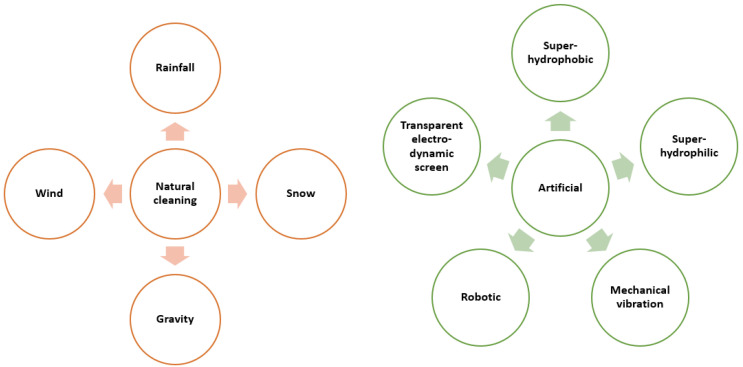
Different cleaning methods for removing dust from solar collectors. Adapted from Ref. [58].

**Figure 7 materials-15-07139-f007:**
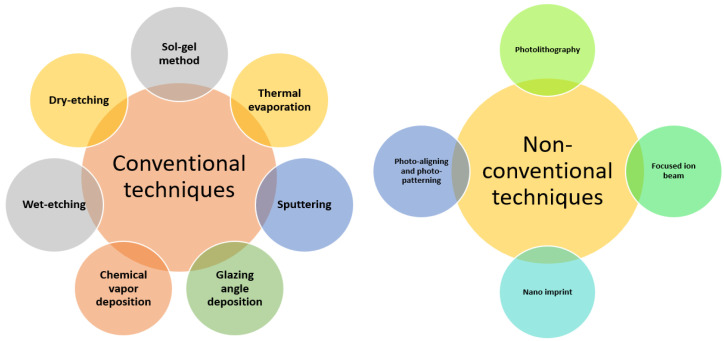
Classification of commonly used ARC fabrication techniques.

**Figure 8 materials-15-07139-f008:**
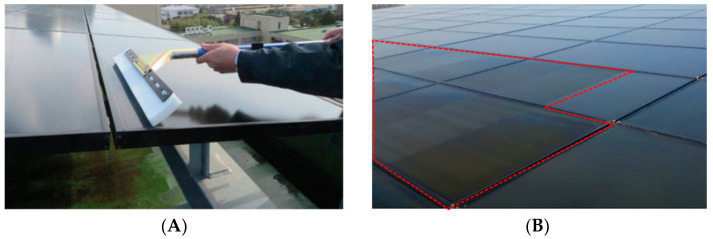
(**A**) Coating method using sponge phase resin; (**B**) surface of the PV modules [57].

**Figure 9 materials-15-07139-f009:**
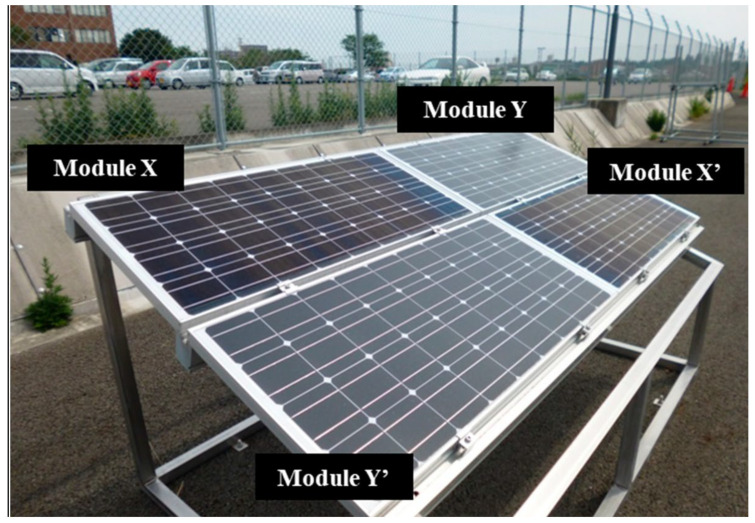
The surface of silicon PV modules. Two modules were coated (modules X and X’) and the other two modules were not coated to act as references (modules Y and Y’) [57].

**Figure 10 materials-15-07139-f010:**
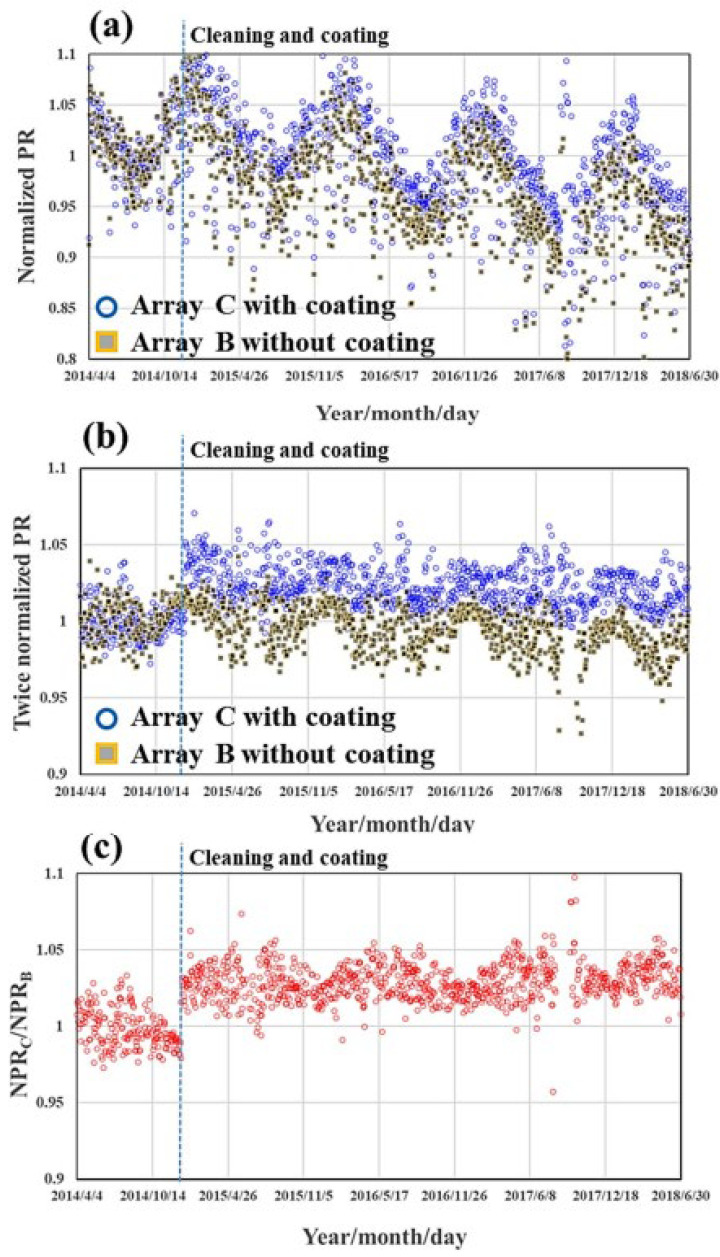
(**a**) Normalized daily performance ratio (PR) of arrays B (NPRB) and C (NPRC), (**b**) twice normalized daily PR of arrays B and C (NPRB and NPRC were divided by the normalized daily PR of reference array A), and (**c**) NPRC/NPRB [57].

**Figure 11 materials-15-07139-f011:**
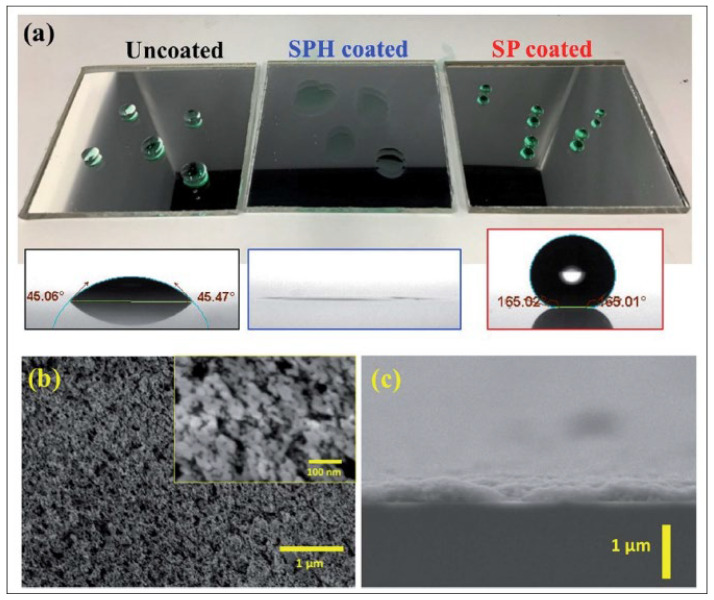
(**a**) Highly reflective, anti-soiling nanoparticle texture solar mirrors (7.6 × 7.6 cm^2^) with no coating, super-hydrophilic (SPH) coating, and super-hydrophobic (SP) coating. Insets are water contact angle measurements on the coated mirrors (the measured water droplet volume = 5–10 µL). (**b**) SEM images of coated mirror surface, showing characteristics of SPH nanotexturing. (**c**) SEM cross-sectional view of SPH coating [62].

**Figure 12 materials-15-07139-f012:**
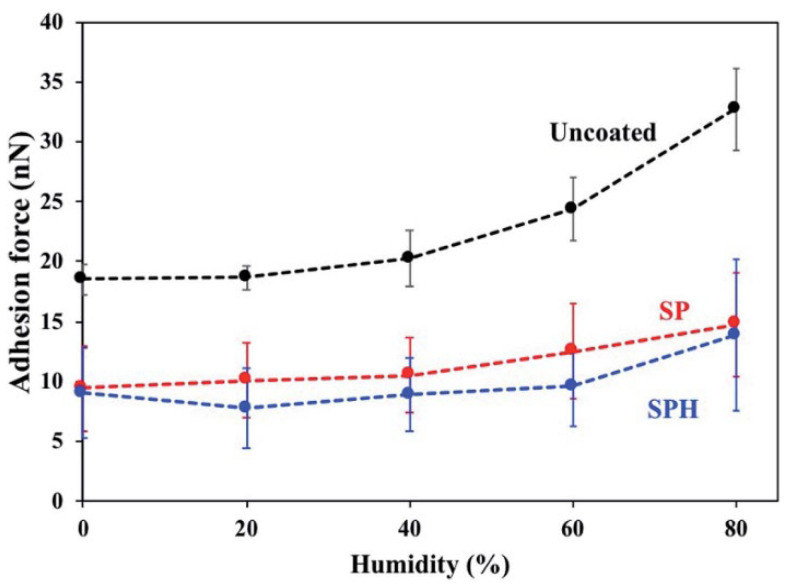
Adhesive force of spherical dust particles with 6 mm diameters on SPH-coated, SP-coated, and uncoated mirrors as a function of relative humidity [62].

**Figure 13 materials-15-07139-f013:**
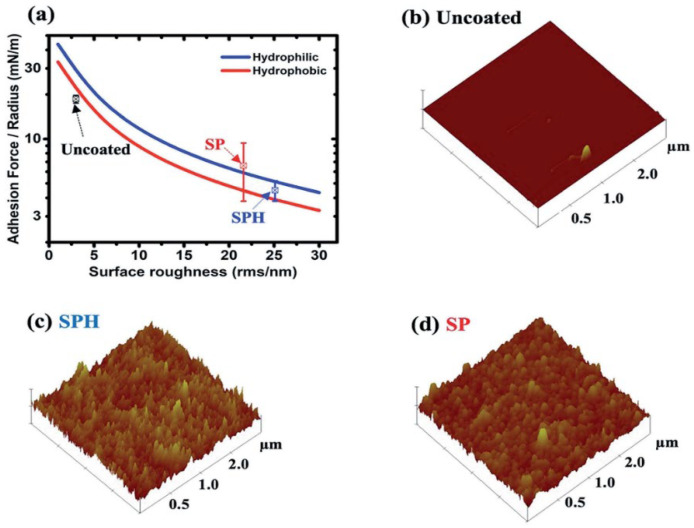
(**a**) Measured and calculated adhesion force between a silica sphere (15 mm diameter) and SP and SPH surfaces. AFM surface morphology analysis of (**b**) uncoated surface, (**c**) SPH coating, and (**d**) SP coating [62].

**Figure 14 materials-15-07139-f014:**
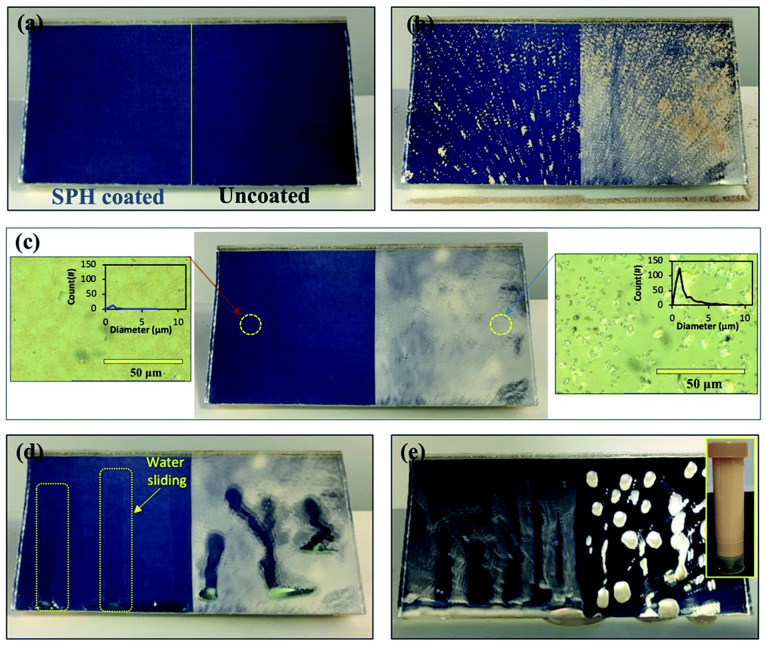
Observation of soiling on a mirror (15.2 × 7.6 cm^2^) with half its area coated with nanoparticles to give it SPH properties. (**a**) A half-coated mirror at 45° elevation, (**b**) initial soiling (1 g) on the mirror, (**c**) dust accumulation on the mirror surface after airbrushing. Insets are optical microscope (5000× magnification) images of soiling on an SPH-coated mirror and an uncoated mirror after airbrushing. (**d**) Water dripping on the mirror surfaces. The water was colored with a green dye. (**e**) Water disperses soil after dripping on the mirror surfaces. Inset is the suspended soil that was collected from the suspension in 0.1 g mL^−1^ of soil and water mixture (ISO 12103-1 A4 Coarse Sand) [62].

**Figure 15 materials-15-07139-f015:**
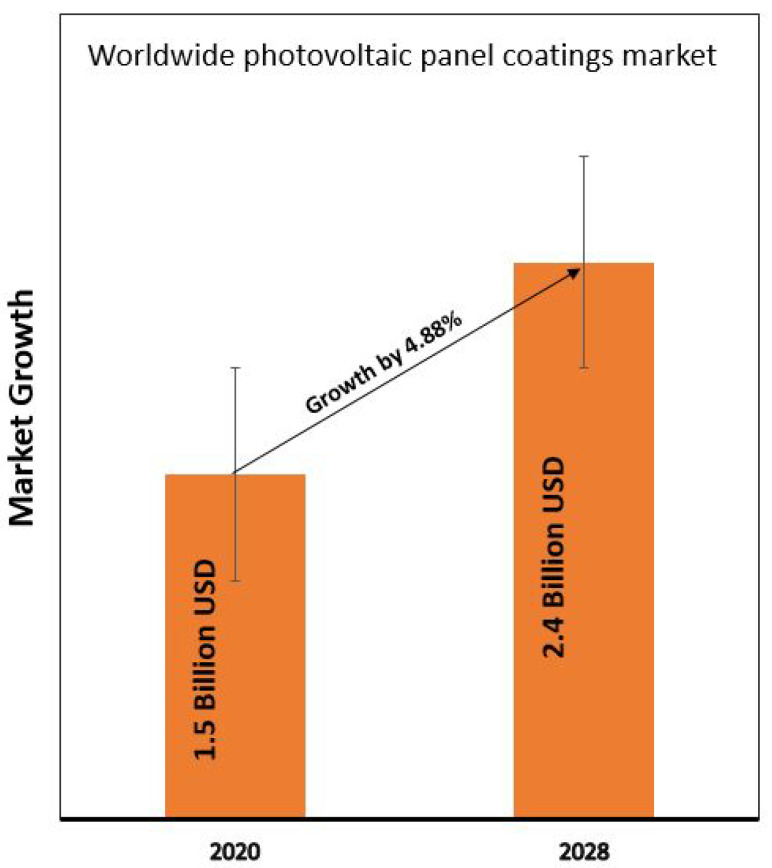
Global solar panel coating market, adapted from [75].

**Figure 16 materials-15-07139-f016:**
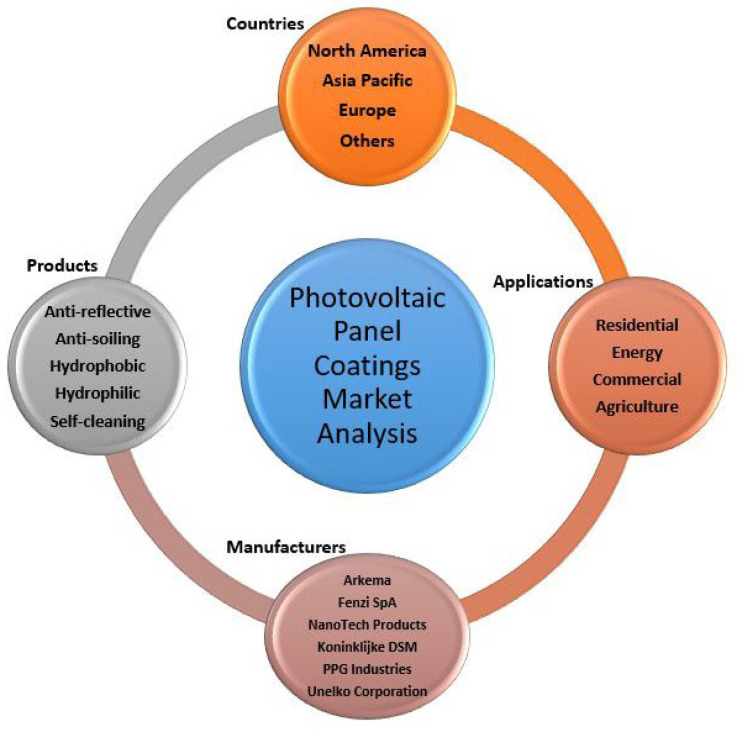
Solar panel coatings market segmentation analysis. Adapted from [76].

**Table 1 materials-15-07139-t001:** Chemical composition of SP and SPH coating determined by XPS [62].

	O–Si	O	Si–O	Si	C	C-Fx	F	O–Si/Si–O	C-Fx/F
Uncoated	53.7	6	27.1	0	7.2	0	0	2.0	—
SPH	53.1	10.0	27.1	3.0	6.3	0	0	2.0	—
SP	45.8	0.4	22.5	0.0	7.7	7.0	15.5	2.0	0.45

**Table 2 materials-15-07139-t002:** Notable achievements in the development of TiO_2_-based anti-reflective coatings (ARCs) and figures of merit showing the resulting enhancement in solar cells’ performance.

Cell Type	Coating Method	ARC Materials	ARC Thickness (nm)	Reflectivity Measurement		Cell Characteristics				
				Range, λ (nm)	Reflectance (%)	J_sc_	V_oc_	FF	ɳ (%)	
Dye-sensitized solar cell (ITO/p-TiO_2_)	Without ARC RF magnetron	TiO_2_ sputtering		-	-	6.17	0.662	6.17	2.30	[65]
			60	-		8.93	0.669	0.58	3.44	
Multicrystalline silicon solar cells GaAs solar cell	Without ARC APCVD	TiO_2_ without ARC electron beam TiO_2_ evaporation and dry TiO_2_ (SWS) etching		300–1150	35.0	28.63	0.561	70	11.24	[66]
			60	8.6		33.86	0.585	72	14.26	
				350–900	>40	18.38	1.011	79.22	14.74	[67]
			∼63	9.5		23.65	1.012	79.46	18.98	
			50	6.2		24.82	1.012	78.42	19.66	
Monocrystalline silicon solar cell	Without ARC sol–gel process and TiO_2_ dip coating SiO_2_/TiO_2_ (DLAR) SiO_2_/SiO_2_- TiO_2_/TiO_2_ (TLAR)			400–1000	37.0	16.2	0.61	76.4	11.36	[68]
			56.8	9.3		24.8	0.61	76.2	14.49	
			41.3/64.6	6.2		25.8	0.61	76.6	14.99	
			56.8/	3.2		27.1	0.61	76.7	15.85	
			69.4/86.8							
Silicon solar cell	Without ARC DC reactive	TiO_2_ magnetron SiO_2_/TiO_2_ sputtering (DLAR)		400–900	36.0	24.8	0.62	79.1	12.2	[69]
			∼100	10.3		34.2	0.63	77.9	16.8	
			∼90/∼110	3.7		37.2	0.63	78.2	18.4	
Monocrystalline silicon solar cell	Without ARC RF magnetron	SiO_2_ sputtering TiO_2_/SiO_2_ (DLAR)		400–1000	35	9.24	0.442	69	2.8	[61]
			81.1	15		12.35	0.504	72	4.5	
			18.0/40.7	7		16.13	0.520	75	6.2	
CIGS: Cu(In,Ga)Se_2_ solar cell	Without ARC Sol–gel process and TiO_2_–SiO_2_ dip coating	Stacks		250–2500	-	33.36	0.510	65.85	11.20	[70]
			Six stacks	-		35.12	0.521	66.40	12.15	
			36–115							
Crystalline silicon solar cell	Without ARC electron beam evaporation	TiO_2_ In/TiO_2_ Al_2_O_3_/In/TiO_2_		400–1050	-	26.10	0.54	-	10.96	[71]
			20	-		30.38	0.55	-	12.84	
			3.8/20	-		32.16	0.55	-	13.69	
			65/3.8/20	-		39.89	0.55	-	16.93	

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
