# Peer review of "Anti-Soiling Coatings for Enhancement of PV Panel Performance in Desert Environment: A Critical Review and Market Overview"

_materials, 2022, doi:10.3390/ma15207139_

Round 1

Reviewer 1 Report

Research on the anti-soiling coating of PV modules is a very interesting research topic. In the draft, the authors summarised the work principle and materials selection for anti-soiling coatings with critical comments. The draft can be accepted after adding more information:

1. For part 4, there is only one subtitle, no need to put 4.1.  For the superhydrophobic coating, what kind of research has been done to improve the durability of superhydrophobic coating?

2. For part6 , the authors should add the working principle of self-cleaning coating and summarize generally used materials for self-cleaning coating.

3.  Will the icing formation affect the solar module performance in winter in certain area?

Author Response

1- Thank you so much. We have already removed the subsection 4.1 and a new paragraph has been added, showing the approaches adopted for improving the durability (and reliability) of such layers: “Until now, various approaches have been investigated to improve the durability of super-hydrophobic surfaces.  Such approaches include the growth of SiO2 layers with nano-particles, low surface energy substances, polymers, intertwined cellulose/SiO2 layers, and optimized micro/nano- hierarchical structure etc [72-74].”

2- Agree. The working principle and used materials have been added in section 6 as requested.

3- We fully agree with you:  icing can be a serious issue especially in some parts of the world. We have added a paragraph accordingly to highlight this point: “In particular regions, snow can act as a degrading factor for output performance of photovoltaic panels. It is considered as a highly reflective material, which can reduce the incident solar radiation and may result in performance drops of the photovoltaic panels. Both super-hydrophobic and super-hydrophilic surface have been suggested as solution to ice formation along with surface heating, however, till now no solid ice-phobic surfaces have been demonstrated.”

Reviewer 2 Report

In this work, the authors reviewed the development of Anti-Soiling Coating techniques, which is critical for PV panel commercial application. As the authors concluded, “…, the overall performance benefit is higher for a hydrophilic surface in high humid areas (coastal) and a hydrophobic surface is dry areas (central desert).” In a brief summary, the review was organised in a smooth flow and included the basic anti-soiling coating technique development in the community. The work can be considered to be published with revisions.

1.       Please use scientific language. For example, Line 86, “very inexpensive” can be replaced by “highly economic” or “highly cost-saving”.

2.       Line 105, the authors used “Lately”. However, the all literatures [25-30] cited was published before 2015, which is 7 years ago. Please consider it.

3.       The references need to be checked. From ref. 57, the sequence of references is not correct. There is no [58] and after [57], it is [61] and then [69]. The following references need to be checked one by one.

4.        Line 369-370, “it is believed that super-hydrophilic surfaces will be the best competitors for developing anti-soiling coatings” looks as a contradictory statement in the whole review, especially in a desert environment?

5.        Line 412, “the market research is driven by four important properties such as anti-reflective, hydrophobic, self-Cleaning, anti-Soiling, and anti-Abrasion.”  It is “four properties” but with five items included in the sentence. Please check and correct.

Author Response

1- Thank you so much. The word has been changed with “highly economic” as suggested.

2- True, thank you. “Lately” has been replaced by “In previous works”. 

3- Thank you so much. The references have been checked and corrected accordingly. 

4- We fully agree with you and the sentence has been removed as suggested.

5- Agree. "hydrophobic" has been removed as it falls under anti-Soiling coating. 

Reviewer 3 Report

The paper mainly discusses the anti-soiling technique for photovoltaics installed in desert areas. Considering the potential energy generation from photovoltaics in the MENA area, the paper is very timely and helpful. Especially, the summary of the anti-soiling market allows us to understand the current situation more clearly. However, the following issues should be addressed before its publication.

Q1: According to Figure 1 and other comments, high humidity and/or dew is required for the process of cemented particles. However, the desert is a very dry atmosphere. Is it a representative model to explain the build-up of dust on the PV module installed in the MENA region?

 Q2: The author well summarized previous works in Table 2. However, most of the works introduced in the table are conducted in moderate (China) and tropical (Malaysia) climate zones. However, the desert area is a dry land with relatively low humidity. In such a climate zone, the effect of dust and its accumulation would be different from those of previous works. Please comment on the difference between previous work conducted under high humidity and in a desert area.   

Q3: In lines 420-425, the author addresses some companies that develop anti-coating layers. First of all, the format does not follow the journal's guidelines. Second, why do you introduce these companies? Are there any reasons for addressing them?  

Author Response

1- Teh desert is hot, sunny dusty but very humid climate. There are some dry deserts as in Chile, but the one characterizing MENA region is very humid (more in certain season that others).

2- We fully agree with you. The paragraph has been modified as follows: “Though the dust condensation is relatively higher in the MENA region, moisture effects take place as studied previously [13,14,15].”

3- New references have been added starting from 2020. The commercial compagnies were introduced as they are the leaders in ADC coatings commercialization, which is the context of this review.

Reviewer 4 Report

The manuscript is appropriate for publishing.

Author Response

Thank you so much for your valuable time and efforts.

Reviewer 5 Report

The paper focused on an interesting topic which is anti-soiling coating.

There are some issues that needs to be fixed:

1.     The statements are mainly general and needs to be more specific in terms of comparisons, strengths, and weaknesses.

2.     The methods for ARC are mentioned, however, there are no details on the comparison between them (cost, availability, robustness, etc)

3.     Some of the paragraphs are not well-placed or not completed. For example, on pages 6, 182-183, it is necessary to be extended substantially.

4.     There are several statements without any reference to support. For example, on page 8, lines 220-222.

5.     English style needs to be checked. Do some statements need to be rewritten, for example, on page 2, line 79?

Author Response

1- Thank you so much. Few paragraphs and statements have been added to define more specific technical comparisons and drawbacks.

2- We fully agree with you. Actually, table 2 complies the performance of ARC films and a new paragraph below has been added to address the robustness.

3- We agree with you and these paragraphs have been added accordingly for a better and more fluent reading. The MS has been proofread at large. 

4- References have been added and cited accordingly for such statements.

5- Again, we agree with you and the sentence has been rewritten as suggested.

Round 2

Reviewer 5 Report

The authors have modified the manuscript and the changes are sufficient. I suggest the manuscript for publication.